# Effect of Chemically Modified Carbon-Coated Iron Nanoparticles on the Structure of Human Atherosclerotic Plaques Ex Vivo and on Adipose Tissue in Chronic Experiment In Vivo

**DOI:** 10.3390/ijms23158241

**Published:** 2022-07-26

**Authors:** Shamil Akhmedov, Sergey Afanasyev, Natalia Beshchasna, Marina Trusova, Ivan Stepanov, Mariya Rebenkova, Ekaterina Poletykina, Yuri Vecherskiy, Sergei Tverdokhlebov, Evgeny Bolbasov, Sascha Balakin, Joerg Opitz, Anatoly Yermakov, Boris Kozlov

**Affiliations:** 1Cardiology Research Institute, Tomsk National Research Medical Center, Russian Academy of Sciences, 634012 Tomsk, Russia; tursky@cardio-tomsk.ru (S.A.); i_v_stepanov@mail.ru (I.S.); mariambf@mail.ru (M.R.); vjj@cardio-tomsk.ru (Y.V.); bnkozlov@yandex.ru (B.K.); 2Fraunhofer Institute for Ceramic Technologies and Systems IKTS, 01109 Dresden, Germany; sascha.balakin@ikts.fraunhofer.de (S.B.); joerg.opitz@ikts.fraunhofer.de (J.O.); 3Research School of Chemistry and Applied Biomedical Sciences, Tomsk Polytechnic University, 634050 Tomsk, Russia; trusova@tpu.ru (M.T.); eyp11@tpu.ru (E.P.); 4School of Nuclear Science and Engineering, Tomsk Polytechnic University, 634050 Tomsk, Russia; tverd@tpu.ru (S.T.); ebolbasov@gmail.com (E.B.); 5M.N. Miheev Institute of Metal Physics of Ural Branch of Russian Academy of Sciences, 620108 Ekaterinburg, Russia; yermakov.anatoly@gmail.com

**Keywords:** atherosclerosis, atherosclerotic plaque, chemically modified carbon-coated iron nanoparticle, reverse cholesterol transport, macrophage, coronary stent

## Abstract

The high mortality rate caused by atherosclerosis makes it necessary to constantly search for new and better treatments. In previous reports, chemically modified carbon-coated iron nanoparticles (Fe@C NPs) have been demonstrated a high biocompatibility and promising anti-plaque properties. To further investigate these effects, the interaction of these nanoparticles with the adipose tissue of Wistar rats (in vivo) and human atherosclerotic plaques (ex vivo) was studied. For the in vivo study, cobalt–chromium (CoCr) alloy tubes, which are used for coronary stent manufacturing, were prepared with a coating of polylactic acid (PLA) which contained either modified or non-modified Fe@C NPs in a 5% by weight concentration. The tubes were implanted into an area of subcutaneous fat in Wistar rats, where changes in the histological structure and functional properties of the surrounding tissue were observed in the case of coatings modified with Fe@C NPs. For the ex vivo study, freshly explanted human atherosclerotic plaques were treated in the physiological solution with doses of modified Fe@C NPs, with mass equal to 5% or 25% relative to the plaques. This treatment resulted in the release of cholesterol-like compounds from the surface of the plaques into the solution, thus proving a pronounced destructive effect on the plaque structure. Chemically modified Fe@C NPs, when used as an anti-atherosclerosis agent, were able to activate the activity of macrophages, which could lead to the destruction of atherosclerotic plaques structures. These findings could prove the fabrication of next-generation vascular stents with built-in anti-atherosclerotic agents.

## 1. Introduction

The high mortality rate caused by cardiovascular diseases, particularly by atherosclerosis, makes it necessary to constantly search for new principles of active influence on the formation of atherosclerosis. Taking into consideration that the growth of atherosclerotic plaque in the vascular lumen occurs at the cellular/molecular level and is accompanied by a chronic inflammatory condition, a targeted influence of the plaque’s molecular and cellular components using nanomaterials offers a promising solution.

The initiating event in atherogenesis is the accumulation of lipoprotein particles within the arterial wall. The maladaptive local effects of particle retention entail not only the aggregation of those lipid molecules contained in the particles, but also the prolonged exposure of these particles to local enzymes and other factors inside the wall of the vessel. This increases their retention, allowing macrophages and vascular smooth muscle cells to take up the protein-lipid material, contributing to the development of foam cells and, ultimately, the growth of atherosclerotic plaques [1].

High-density lipoproteins (HDL), composed of apolipoprotein A1 (apoA1) and phospholipids surrounding a core of cholesteryl esters and triglycerides, are responsible for removing excess cellular cholesterol from the peripheral tissues and transporting it back to the liver. This atheroprotective process is called reverse cholesterol transport (RCT) [2]. RCT is believed to be a protection mechanism against cardiovascular diseases. Increasing HDL levels offers a promising therapeutic strategy for preventing, and possibly reversing, atherosclerosis through increasing RCT [3].

While lipid-lowering drugs have been successfully introduced over the last 25 years, new therapeutic techniques that concentrate either on treating vessel wall inflammation or inducing local therapeutic effects on plaque macrophages have been explored in more recent years [4]. Current research has also focused on improving stent coatings, which will enable stents to serve as vectors for local drug distribution and function as inductive scaffolds for plaque destruction and endothelial repair. The integration of micro and nanotechnology into stent architecture could be used to increase the versatility of local drug release, and show the potential for targeted therapy at the immediate location of implanted stents.

Different superparamagnetic iron oxide nanoparticles have been investigated for their impacts on the formation of atherosclerosis. Although mainly used for atherosclerotic plaque imaging [5], they have also been applied for therapeutic purposes in atherosclerosis, e.g., by reducing the plaque hardness through heating of the plaque using an external magnetic field [6] or magnetic drug targeting [7].

To reduce atherosclerotic lesion formation, liposome-like nanoparticles based on phosphatidylcholine and a CD36-targeting ligand have been developed [8]. Such nanoparticles reduce lesion formation by blocking specific membrane receptors of intimal macrophages, a.k.a foam cells, which ultimately decreases lipid accumulation and inflammation [9]. Moreover, the targeting nanoparticles have been studied as drug carriers to deliver siRNA to reduce the number of inflammatory monocytes, or to deliver LXR agonists which increase cholesterol efflux [10,11]. Current strategies are focused on the targeting of macrophages. It would theoretically be possible to use not only Fe@C NPs for the goals and objectives outlined in this article, but also other nanoparticles made of nickel, palladium, and other metals [12]. However, according to the literature, some types of nanoparticles have a very high toxicity to biological tissues. Therefore, when discussing a strategy for choosing the optimal nanoparticles, a basis type of nanoparticles of zero-valent iron, which, according to the own data of the authors, had the lowest toxicity, was favored. The uniform coating of iron nanoparticles with a carbon layer of about 5 nm reduced the toxicity and, simultaneously with the help of diazonium tosylate salts, formed lipophilic organic groups on their surface. The modified Fe@C NPs were placed into contact with atherosclerotic plaques and changed their structures [13].

This work continues the preliminary work of the authors where the mechanisms of synthesis and functionalization of Fe@C NPs with lipophilic organic groups, which mimic atheroprotective biomimetic HDL, were studied, and their physical–chemical behavior in vitro and in vivo was characterized [13,14,15]. In those works, native human atherosclerotic plaques were obtained during open carotid endarterectomy surgery, treated ex vivo with Fe@C NPs, and implanted into Wistar rats. The authors were able to show that, based on histomorphological and electron-microscopic studies, the Fe@C NPs penetrated the inner structures of the plaques and caused significant structural changes, depending on the period of implantation. The high biocompatibility of Fe@C NPs demonstrated in these preliminary works reinforces the hypothesis that Fe@C NPs have a high potential for the treatment of atherosclerosis disease by preventing restenosis and the further formation of atherosclerotic plaques. The current study extends this work by demonstrating the interaction of Fe@C NPs with the adipose tissue of Wistar rats in vivo and human atherosclerotic plaques ex vivo.

## 2. Results

### 2.1. Morphological Study

Morphological studies were carried out to investigate the in vivo interactions of implanted CoCr tubes with PLA and Fe@C NP coatings, as shown in Figure 1. The first sample group contained chemically modified Fe@C NPs in the PLA coating, whereas the second group of samples contained PLA with non-modified Fe@C NPs.

After seven days of implantation in Wistar rats, the morphological investigation showed a non-specific productive inflammation with the formation of non-fibrous connective tissue of laboratory animals in group 1 (Figure 1a). The minimal infiltrate consisted mainly of lymphocytes and macrophages, where some macrophages had black granular inclusions in their cytoplasm. A moderately pronounced infiltration by lymphocytes and macrophages around the agglomerates of the Fe@C NPs, resulting in color and optical density changes of the peripheral zone and manifested by the “fringing” phenomenon (Figure 1a; indicated by red arrows), was observed. A capsule of soft-fibrous connective tissue was formed. Group 2 showed a more pronounced productive inflammatory reaction, with the formation of fields of roughly fibrous connective tissue around the unmodified Fe@C NPs and in the fatty cell periphery. Few macrophages with inclusions of black granular inclusion in the cytoplasm were detected (indicated by yellow arrows in Figure 1b). No fringing phenomenon around the unmodified Fe@C NPs was visible.

After 14 days a pronounced, non-specific chronic productive inflammation was observed around the clusters of modified Fe@C NPs in group 1 (Figure 1c). Additionally, the formation of a connective tissue capsule was found. Non-specific productive inflammation, with pronounced fibrosis spreading into the surrounding fatty tissue and a moderate macrophage infiltration, was observed for group 2 (Figure 1d).

21 days after implantation, a chronic non-specific productive reaction and extensive infiltration by lymphocytes and macrophages were recorded around the clusters of modified Fe@C NPs (Figure 1e). A strong decrease in optical density around the peripheral zones of the Fe@C NPs (fringing phenomenon) with the appearance of a prominent yellow-brown zone was observed (indicated by red arrows in Figure 1e). At the same time, the fibrous transformation of the adipose tissue around the areas with implanted unmodified Fe@C NPs was exhibited by group 2. Furthermore, neoangiogenesis (indicated by yellow arrows in Figure 1f) and an uptake of Fe@C NPs clusters via macrophage phagocytosis was observed in the peripheral zones.

To assess the area of fringing regions around the Fe@C NPs clusters as a function of their contact time with the fatty tissue of laboratory animals, a comparative analysis via optical microscopy was performed. Significant linear growth of the area exhibiting the fringing phenomenon was observed around the modified Fe@C NPs clusters, as shown in Table 1. Figure 2 depicts the correlated histological analysis of the fringing phenomenon (fringing is indicated by red arrows in Figure 2).

### 2.2. Study of the Interaction of Atherosclerotic Plaques with Fe@C Nanocomposites

The dynamics of the Fe@C NP-driven release of cholesterol-like substances from the atherosclerotic plaque into the 0.9% NaCl physiological solution was characterized using chromatography (Figure 3). The study was conducted on 5 different plaques using modified and non-modified Fe@C NPs, each using two different doses (5 wt% and 25 wt%), and one control without the use of NPs. A sharp increase in the concentration of cholesterol-like substances in saline was noted during the first hours of the experiment, after which the levels of released cholesterol-like substances reached a plateau and remained stable throughout the entire observation period. The highest concentration of cholesterol-like substances was observed in the case of modified Fe@C-C18 NPs. The main rationale for this observed phenomenon is that the surface of the NPs is closest to the atherosclerotic plaque structure, which allows the lipophilic tail to penetrate and destroy it [8].

## 3. Discussion

This article does not cover the treatment of atherosclerosis. It is aimed at finding methods and means of influencing the atherosclerotic plaque itself, to find mechanisms that could affect its growth when it is in the lumen of a blood vessel. For this purpose, two experimental models were chosen. One model was aimed at studying the interaction of the adipose tissues of an experimental animal with chemically modified Fe@C NPs, because adipose tissue is more involved in the formation of atherogenesis. With the help of morphological research methods, the authors were able to show that it is modified Fe@C NPs, and not simply NPs in their pure form, that have the property of a special interaction with the adipose tissues of experimental animals. This phenomenon has been named “bordering phenomenon”. No description of this phenomenon has been found in the existing literature. At the morphological level, this phenomenon can be described as additional accumulations of macrophages in the zone that is located between the adipose tissue and modified Fe@C NPs. However, given that, according to the literature [13], it is macrophages that play a leading role in the processes of influencing atherogenesis, we consider this phenomenon as additional possible evidence for the formation of atherogenesis in general. The second experimental model of the presented article was complementary to the first model. Based on the example of a live atherosclerotic plaque, which was obtained during a planned operation of carotid endarterectomy, it was shown that the plaque is affected by modified Fe@C NPs. This phenomenon was confirmed by the significantly greater release of cholesterol-like substances into the physiological solution compared with the control and untreated Fe@C NPs. Moreover, the yield of these substances directly depends on the applied concentration of modified Fe@C-C18. At 25 wt% concentration of modified Fe@C NPs, the level of release of cholesterol-like substances into the physiological solution is three times higher than at 5 wt% concentration.

Due to the excellent anti-atherosclerotic properties of native HDL molecules, different approaches for the fabrication of HDL-mimicking NPs have been reported in recent years with a good success. A high ability of HDL-like NPs to target atherosclerotic plaques can be explained by their affinity to particular types of immune cells, namely monocytes and macrophages. However, the mechanism of interaction between HDL and plaques is still poorly understood.

This work is a logical continuation of our previously published material [8,9,10]. In this study, Fe@C NPs were obtained by the covalent modification of the surfaces of iron metal nanoparticles coated with carbon by applying aryldiazonium tosylates. The presence of lipophilic organic reactive groups on the surfaces of the modified Fe@C NPs served as a drug-free biomimetic substitute for atheroprotective HDL.

As a result of a comprehensive morphological study, clusters of modified Fe@C NPs were more susceptible to change in their structure than clusters of non-modified Fe@C NPs. The organic nature of the modified Fe@C NPs caused a more pronounced tissue nonspecific reaction, characterized by the activation of the macrophage link of phagocytosis, increased lysis activity of tissue-specific enzyme systems, and certain changes in the cell-stromal interactions. Obviously, the modified nanoparticles changed the histological structure and functional properties of the surrounding adipose tissue of laboratory animals. Compared with other types of NPs, which need to use drugs or targeting ligands to achieve therapeutic efficacy for atherosclerosis treatment and diagnosis, respectively, the applied Fe@C NPs demonstrate intrinsic anti-atherosclerotic activity and targeting ability for atherosclerotic plaques.

A natural nanoparticle, HDL is known to possess a capability to attract macrophages of atherosclerotic plaques and promote an efflux of cholesterol from cells. Although the targeted nanotherapeutics have not been yet applied for atherosclerosis therapy, they have shown promising effects in pre-clinical models related to their ability to modulate plaque macrophage functions.

The present study aimed to understand the influence of spherically symmetric iron nanoparticles encapsulated in carbon on adipose tissue in a chronic experiment in vivo and on the structure of human atherosclerotic plaques ex vivo. Fe@C NPs, non-modified and modified with diazonium salts, were subjected to phagocytosis by macrophage cells in an in vivo experiment. Clusters of nanoparticles underwent changes, which were manifested by the transformation of color and optical density around the nanoparticles—referred to as the “fringing phenomenon”. The described changes were most likely related to the phagocytic and enzymatic activity of macrophages, synthesis of biologically active substances, and several other yet-poorly-studied mechanisms of intercellular interactions.

The modified Fe@C NPs were more susceptible to changes of their colour and optical properties compared with their non-modified analogues. These changes increased with the accumulation of implantation time. The modified Fe@C NPs have shown a more pronounced efflux of cholesterol-like substances from atherosclerotic plaques exposed to physiological solution in the ex vivo experiment.

All the phenomena described in this article, ranging from morphological evidence of the interaction of modified Fe@C NPs with adipose tissues of experimental animals to the results obtained when studying their interaction with living human plaques, show promise in the application of nanoparticles for their effect on the mechanisms of atherosclerosis formation in the blood vessels.

## 4. Materials and Methods

### 4.1. Samples for Chronic Experiment In Vivo

Co–Cr metallic tubes for the manufacturing of bare metal stents, 2–3 mm in length and 1.5 mm in diameter, were coated 95 wt% of polylactic acid (PLA, PURASORB^®^ PL 10, Corbion, Amsterdam, The Netherlands) mixed with either modified Fe@C nanoparticles or non-modified Fe@C nanoparticles, each at a concentration of 5 wt%. The carbon-coated iron nanoparticles Fe@C with a size of ~10 nm were obtained by gas-condensation synthesis and modified with 4 octadecyl benzene diazonium tosylate and lipophilic organic groups, which mimic atheroprotective biomimetic HDL [13,14,15]. The technology of Fe@C NP preparation and chemical modification was described in the previous works of the authors [16,17]. The coatings were formed using a homogeneous solution of polylactic acid in trichloromethane with the addition of modified Fe@C nanoparticles. The formation of a composite coating on the surface of the cylindrical specimens was carried out by aerodynamic molding in a turbulent gas flow using Minijet 4 sprayers (Sata, Kornwestheim, Germany). The application of the composite coating was carried out in such a way as to ensure the homogeneous thickness of the coating within 10 μm. To increase the adhesion of the composite material to the steel substrate, the surface was sequentially washed in trichloromethane (CHCl_3_, 99%, Panreac, Barcelona, Spain) and ethanol (C_2_H_5_OH, 99.8%, Panreac, Spain) before application. The thermal treatment of the formed coatings was carried out using a ITM 50.1100 chamber furnace (ITM, St. Petersburg, Russia) with heating up to a temperature of 200 °C.

### 4.2. Research on Laboratory Animals

The experimental procedure was approved by the Home Office for Care and Use of Laboratory Animals of the Research Institute of Cardiology, Tomsk National Research Medical Center—Russian Academy of Science (Corresponding ethical approval protocol no. 112 from 23 October 2013), and performed with a with a strict adherence to the WMA Statement on Animal Use in Biomedical Research (2006).

Thirty adults male Wistar rats, aged between 12 and 15 weeks, weighing between 220 and 250 g, fasted for 12 h, and were allowed free access to water before invasive interventions. The animals were housed at 20 °C and offered rat chow and water ad libitum. They were kept in dark: light cycles (DL = 12:12 h) in individual wire-bottomed cages. Two groups of animals were formed (n = 15 each). Into the subcutaneous fat of the first group of animals (n = 15), Co–Cr tubes coated with PLA containing modified Fe@C NPs were implanted under general anesthesia (Zoletil-100) (Figure 4a). The second group of animals (n = 15) was equipped with CoCr tubes whose PLA coating contained non-modified Fe@C NPs. Each group was divided into two subgroups of 5 animals, depending on the time of animals’ withdrawal from the experiment. Surgical operations for the explantation of the CoCr tubes were performed after 1, 2, and 3 weeks, respectively. The implant was excised along with the surrounding soft tissues (Figure 4b) and was placed in a 10% buffered formalin solution for further study.

All manipulations were performed in sterile conditions, and following surgery the animals were kept in standard vivarium conditions with free access to water and food. The purpose of this study was to evaluate morphofunctional reactions at the interface between implanted CoCr tubes coated with PLA vs. Fe@C NPs and biological tissues of experimental animals.

### 4.3. Morphological Studies

To evaluate possible toxic effects of the implanted CoCr tubes containing modified and non-modified Fe@C NPs on the biological tissues of the laboratory animals, the investigated tissue samples were examined for morphofunctional abnormalities via histology. Tissue samples were dehydrated in a solution for histological processing based on absolute isopropyl alcohol ISOPREP^®^ and impregnated in a homogenized paraffin medium using an automatic histological Excelsior™ AS Tissue Processor (Thermo Fisher Scientific, Waltham, MA, USA) and a paraffin filling station Tissue-Tec^®^TEC™ 5 EM E-2 5230 (Sacura, Osaka, Japan). Paraffin sections with a thickness of 5–7 µm were created using a rotary mechanical microtome HM 355S (Thermo Fisher Scientific, Waltham, MA, USA), followed by staining with haematoxylin and eosin on a ST5010 XL staining machine (Leica, Wetzlar, Germany), also by the Van Gieson method. The colored histological tissue samples were placed into a synthetic mounting medium ConsulMount with an automatic coverslipping ClearVue™ machine (Thermo Fisher Scientific, Waltham, MA, USA), in accordance with the instruction manual provided by the supplier. Microphotographs of histological preparations were acquired using an Axiocam 506 color camera (Carl Zeiss, Oberkochen, Germany). The severity of the toxicological effects of implanted CoCr tubes vs. PLA with Fe@C NPs on the biological tissues of laboratory animals on a morphofunctional level was assessed based on the following parameters: content in histological tissue samples of certain cell types (polymorphonuclear cells, lymphocytes, macrophages, and giant multinucleated cells), the severity of neovascularization, fibrosis, and necrosis.

### 4.4. Ex Vivo Experiment with Human Atherosclerotic Plaques

Atherosclerotic plaques were obtained during elective endarterectomy operations in 20 male patients aged 63 ± 1.1 years. These patients had an extended atherosclerotic plaque narrowing the lumen of the internal carotid artery by 75%, according to ultrasound examinations (Figure 5a). None of the patients took cholesterol-lowering drugs on an ongoing basis. All surgeries were performed at the Cardiology Research Institute, Tomsk National Research Medical Center, Russian Academy of Sciences, according to usual protocols.

Immediately after excision, the plaques were placed in a sterile physiological solution (Figure 5b) and transferred to the laboratory, where the sterility regime was observed during all subsequent stages of the study. Each plaque was divided into three fragments. One fragment was the control fragment (fragment I) and the other two plaque fragments were treated at a ratio of 5 and 25 wt% of the amount of atherosclerotic plaque with non-modified Fe@C NPs (fragment II) and modified Fe@C NPs (fragment III), respectively. A plaque was taken, weighed, and 5 or 25% was calculated from the mass of the plaque (for a 1 g plaque: 1 g × 0.05 = 0.05 g nanoparticles or 0.25g, respectively). All three fragments were placed in a 10 mL oxygenated physiological solution at 37 °C and incubated under constant stirring. After 60 min, 1 mL was taken from the incubation medium of each fragment into separate aliquots. These samples were centrifuged at 10,000 rpm and 0.1 mL of each was added to an aliquot containing a mixture of H_2_SO_4_:AcOH:(AcO)_2_O (in a 1:1:5 ratio), giving a colour reaction for cholesterol-like substances. The prepared samples were stored at 34 °C for 25 min and investigated in terms of the presence of cholesterol-like substances by means of a Specord 250 Plus double-beam spectrometer at a wavelength of 625 nm. The reversed phase HPLC analysis was carried out using an Agilent 1260 infinity HPLC system (Agilent, Santa Clara, CA, USA) with a Zorbax Eclipse XDB-C18 (150 mm × 4.6 mm × 5 μm) (Agilent, Santa Clara, CA, USA) column. Chromatographic separation was carried out at a column oven temperature of 40 °C, with a mobile phase flow rate of 1 mL∙min^−1^. The mobile phase consisted of a methanol (HPLC grade, Aquametry) and 0.005 M orthophosphoric acid aqueous solution in an 80/20 ratio. The injected sample volume was 20 μL. The wavelength in a DAD-detector was set at 270 nm.

## 5. Conclusions

The results of both described experiments confirm the potential of chemically modified Fe@C NPs to be used as an HDL-like anti-atherosclerosis agent, able to activate the activity of macrophages, which could lead to the destruction of atherosclerotic plaques structures. By using metallic tubes which are used in stent manufacturing as the carrier for the Fe@C NPs, this work highlights the possibility for producing coronary stents with Fe@C NPs as an anti-atherosclerotic agent in the future.

## 6. Patents

Akhmedov SD, Afanasiev SA, Filimonov VD, Postnikov PS, Trusova ME, Karpov RS. Agent for the selective adjustment of blood lipids. US9,789,134 B2., 17 October 2017.

## Figures and Tables

**Figure 1 ijms-23-08241-f001:**
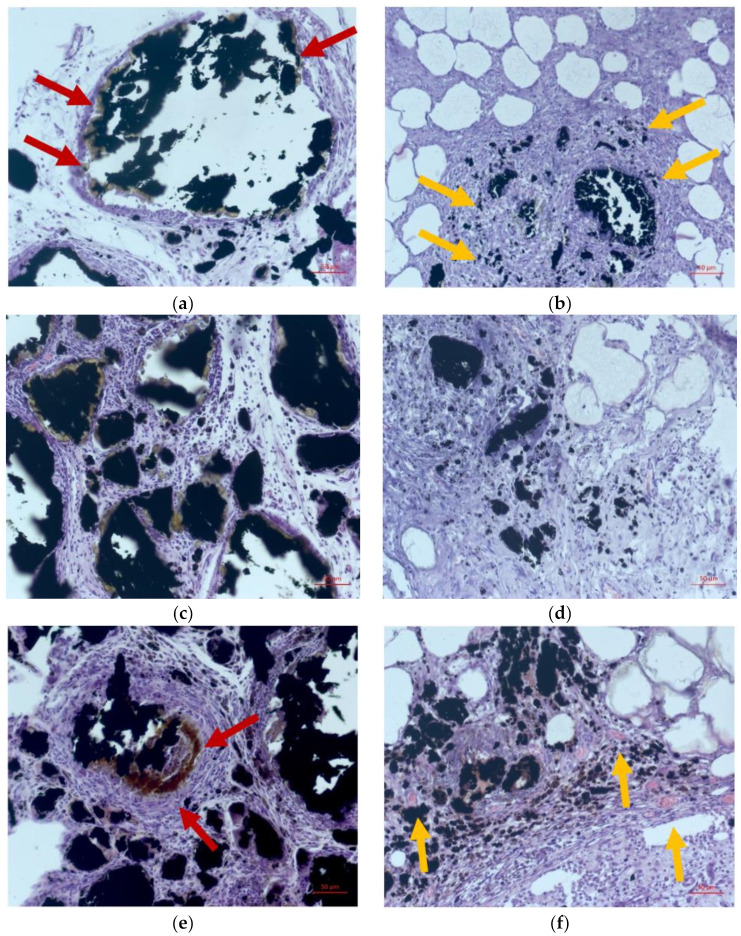
Morphological studies of CoCr tubes with PLA and Fe@C NP coatings after 7, 14, and 21 days of implantation. In samples with modified nanoparticles (group 1) there was a moderately pronounced edema and minimal infiltration by lymphocytes and macrophages around the agglomerates of the nanoparticles. Moreover, it was observed that «fringing» phenomenon manifested by the changes of the color and optical density in nanoparticles peripheral zone. Scale bar is 50 µm. (**a**,**c**,**e**) Modified Fe@C NPs (Group 1). (**b**,**d**,**f**) Unmodified Fe@C NPs (Group 2). Time: 7 days (**a**,**b**), 12 days (**c**,**d**), and 21 days (**e**,**f**). Red arrows indicate the presence of a fringing effect, and yellow arrows indicate its absence.

**Figure 2 ijms-23-08241-f002:**
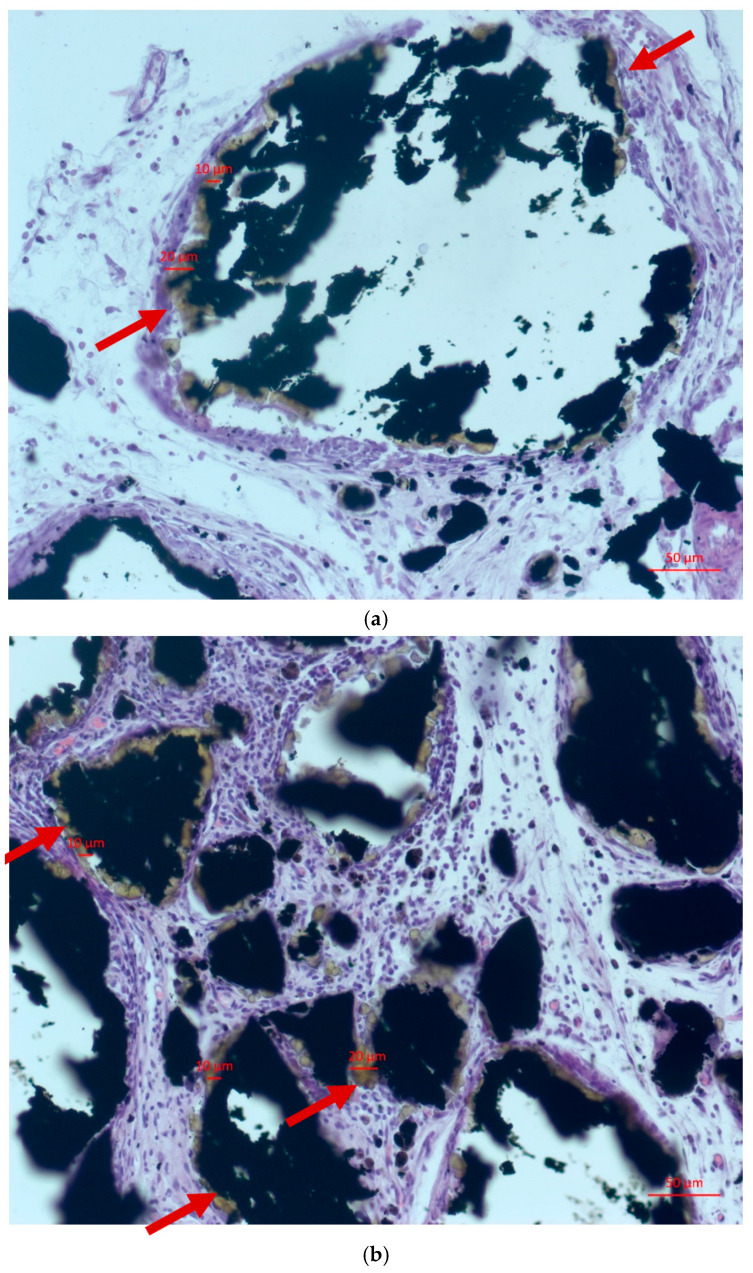
Areas of the fringing phenomenon around the modified Fe@C NPs in the fat tissue of a laboratory animal at 7 (**a**), 14 (**b**), and 21 (**c**) days of the experiment. Red arrows indicate the presence of a fringing effect.

**Figure 3 ijms-23-08241-f003:**
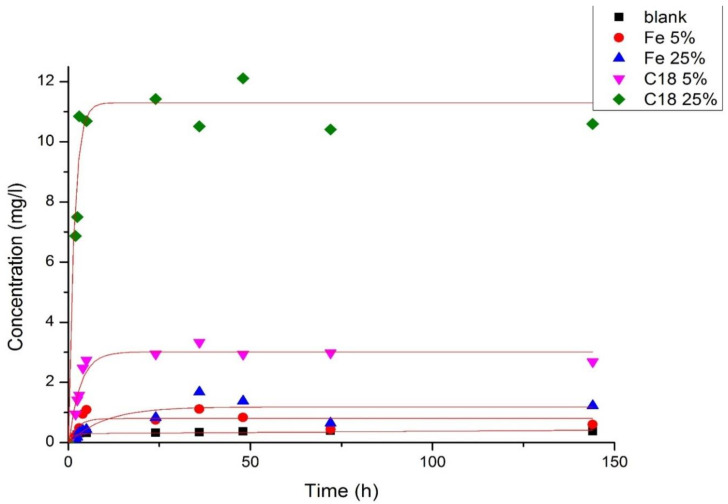
The yield of cholesterol-like substances from the atherosclerotic plaque substance into 0.9% saline under the influence of Fe@C NPs. The black square is the control group. Red circle—non-modified Fe@C NPs at 5% nanoparticle concentration. Blue triangle—non-modified Fe@C NPs at 25% nanoparticle concentration. Pink triangle—modified Fe@C NPs at 5% nanoparticle concentration. Green rhombus—modified Fe@C NPs at 25% concentration of nanoparticles.

**Figure 4 ijms-23-08241-f004:**
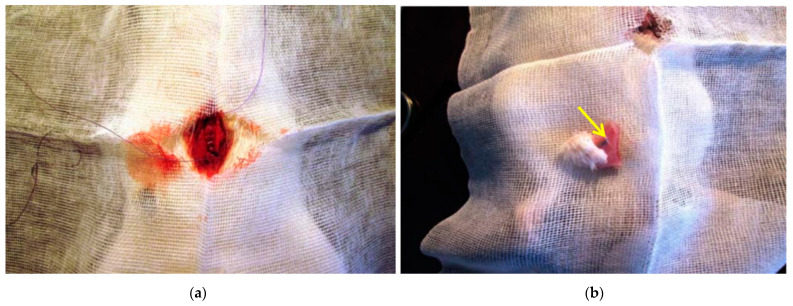
Surgical implantation of CoCr tubes PLA@Fe@C NPs into the subcutaneous tissue of a laboratory animal (**a**); operation of excision of subcutaneous tissue and muscles with a sample metal tube (yellow arrow) coated with a nanocomposite (**b**).

**Figure 5 ijms-23-08241-f005:**
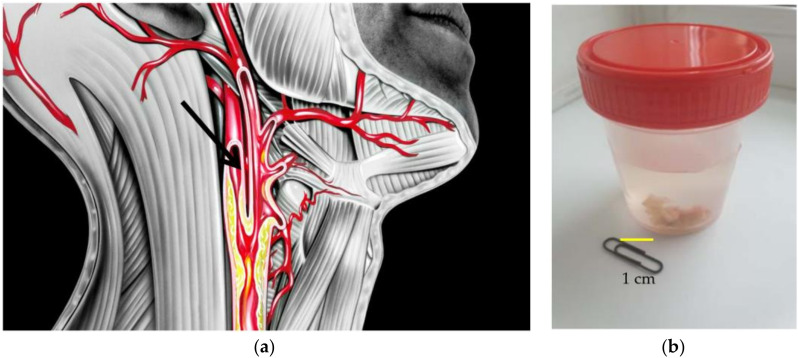
Scheme of the operation of carotid endarterectomy. The black arrow indicates an atherosclerotic plaque that is located in the lumen of the carotid artery. (**a**) and general view of atherosclerotic plaque removed from the carotid artery of the patient and placed into saline solution (**b**).

**Table 1 ijms-23-08241-t001:** Quantitative estimation of metachromatic staining (total area µm^2^) in the structure of the subcutaneous fat of laboratory animals (n = 5, five animals for every group), depending on the time of their contact with chemically modified and non-modified Fe@C NPs.

Duration of Implantation, Days	Area of Fringing Phenomenon, µm^2^	
	Non-Modified Fe@C NPs	Modified Fe@C NPs
7	- *	3.38 ± 1.19
14	- *	6.31 ± 0.10
21	- *	9.62 ± 0.36

* no fringing phenomenon.

## Data Availability

Not applicable.

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
