# Peer review of "Effect of Chemically Modified Carbon-Coated Iron Nanoparticles on the Structure of Human Atherosclerotic Plaques Ex Vivo and on Adipose Tissue in Chronic Experiment In Vivo"

_ijms, 2022, doi:10.3390/ijms23158241_

Round 1
Reviewer 1 Report
In the manuscript by Akhmedov et al, tried to understand the effect of chemically modified carbon-coated iron nanoparticles on structure of human atherosclerotic plaques ex vivo and adipose tissue in chronic experiment in vivo. The manuscript could be improved by addressing following questions and suggestions:
-Please make similar writing format throughout the manuscript (In vivo and Ex vivo should be Italic font)
-What does 5 wt% in abstract (is it w/v?)
-The abstract is not clear (Please rewrite the abstract). What are the results and observations made by authors from the experiments?
-Introduction: Lack of proper citation of the previous work, please arrange the introduction with proper rational and hypothesis of your study based on the previous work.
-Materials and Methods:
-List down all the chemicals used in this study with proper source.
-Please check all the typos: mixed with modified Fe@C nanoparticles ore non-modified Fe@C (in line 79).
-Briefly explain your previous work, how these Fe@C nanoparticles were prepared and their in vitro characteristics.
-There are several typos in the manuscript, please correct.
-Figure 3: In caption put scale bar.
-Table 4: Provide the data as (mean ± SD, number of experimental conditions).
Reviewer 2 Report
The manuscript seems interesting and has been well designed. However, the background study seems poor. I suggest adding ongoing strategies for treating atherosclerosis/plaque, and more recent trends in nanoparticles strategies including carbon-coated iron nanoparticles and other nanoparticles. Also, please discuss how this nanoparticle strategy is better than other ongoing strategies for the treatment of atherosclerosis/plaque.
Round 2
Reviewer 1 Report
Authors have revised the manuscript according to the suggestions and comments, however need to improve in terms of language before published.
Author Response
Thank you very much for your comment. The manuscript is reviewed and corrected in terms of English by English native speaker.
Reviewer 2 Report
The authors have edited the manuscript as per my suggestion.
Author Response
Thank you very much for accepting the manuscript and for your kind review of our work.
This manuscript is a resubmission of an earlier submission. The following is a list of the peer review reports and author responses from that submission.
Round 1
Reviewer 1 Report
Comments are attached.

Author Response
Reviewer I
Comments/Questions:
- What is the weight concentration 5% referring to?
We have included the appropriate explanations in the article.
- Description of groups is confusing in lines 106, 107, 117, 138 and 196. It is
recommended to maintain consistency throughout the manuscript to avoid confusion.
We tried to take into account these wishes in the article.
- In the methods section 2.4 the authors describe treatment of plaques with Fe@C NPs at
a concentration of 10% of the weight of plaques. But in results section 3.2 the
concentration of NPs is mentioned as 5% and 25%. This leads to major ambiguity in
terms of results.
In the specified section 2.4 of the text itself, we have made the appropriate changes and explanations.
- Please use an original figure for figure 2 (left).
We have placed another original picture.
- In figure 3, why are there more black deposits in Modified Fe@C NPs group than the
unmodified group?
Because it is a visual effect that associated with the higher picture magnification in modified Fe@C NPs group. We are need to use this magnification to demostrate „fringing“ phenomenon.
- Correct typographical error in figure 3 legend.
Done.
- In line 233, authors mention extensive infiltration of lymphocytes and macrophages.
However, it is hard to distinguish cells with H&E images provided in the manuscript,
at given magnification. Have the authors performed a quantitative scoring of
inflammation? If so, please describe the procedure in methods section and update the
results.
In present study we assessed the presence of an inflammatory infiltrate as a fact and used a descriptive assessment of inflammation. Quantitative scoring of inflammation are not performed in this study.
Assessment of the cellular composition and typing of inflammantory cells is expected in our further work with using of IHC methods.
- In line 238, authors mention neo-angiogenesis happening in the tissue. Is there any
histological proof that can be provided in the form of CD31 staining comparing the
groups?
The presence of angiogenesis was assessed as the fact of the presence of newly formed blood vessels of the capillary type. The detection of capillaries using CD31 and the calculation of the density of microvessels are planned by us in a further study.
- Table 1 needs major formatting. Labels in column 1 are not clear.
Done by Sascha.
- The importance of fringing phenomenon has not been discussed at all in discussion
section. Adding this bit will help readers correlate the results of this study better.
We tried to take into account these wishes in the article.
- “Fe@C-C18” has not been described earlier in the manuscript.
We have included the appropriate explanations in the article with reference to the literature [8].
- It would help if the authors can provide supporting references for the rationale that they
mention in the lines 276 and 277.
I added our last paper as a reference.
- Discussion section needs a major revision. Mainly, results from documented literature,
that either support or differ from the current research, needs to be discussed to correlate
the results better.
One additional paragraph is added.
- Provide reference for the statement in lines 307 and 308 about HDL.
- In line 315, authors say the NPs were “subjected to phagocytosis”. Phagocytosis is a
natural phenomenon that occurs as part of foreign body response. Unless the authors
have specifically simulated conditions for phagocytosis of NPs in rats, which they
should mention in methods section, it misleads the readers to use the wording
“subjected to phagocytosis”.
We did not pursue the goal of substituting the concept of "Phagocytosis". We wanted to convey the idea that phagocytosis itself, as a natural phenomenon, is actively present in the process under study.
Overall, while the research idea presented in the manuscript seems intriguing, it lacks proper
presentation of results and a healthy discussion which expands on the importance of the results obtained. I would recommend a major revision of the manuscript before it is submitted again
Reviewer 2 Report
In this study, the authors tried to develop functionalized and modified Fe@C nanoparticles aiming to reproduce the atheroprotective properties of native HDL. They have observed in an in vivo experiment that Fe@C NPs modified and non-modified with diazonium salts were subjected to phagocytosis by macrophage and are able to induce fringing phenomenon due to local inflammatory reactions. Moreover, in an ex vivo experiment using fragments of atherosclerotique plaque removed from patients and exposed to modified Fe@C NPs, they have shown a more significant efflux of cholesterol-like substances from atherosclerotic plaques.
Overall, the manuscript is well written.
I do consider that the manuscript may be considered for publication in this journal.
Minor comments:
- I don’t feel that the first images with surgical interventions on rats are really necessary; the same opinion is for the next 2 images showing the carotid endarterectomy and the plaque removed from patient;
- I would suggest to add at least one paragraph in the discussions section in which the authors must emphasize/ discus more about the results obtain from ex vivo experiments and possible mechanism involved.
Author Response
Reviewer II
In this study, the authors tried to develop functionalized and modified Fe@C nanoparticles aiming to reproduce the atheroprotective properties of native HDL. They have observed in an in vivo experiment that Fe@C NPs modified and non-modified with diazonium salts were subjected to phagocytosis by macrophage and are able to induce fringing phenomenon due to local inflammatory reactions. Moreover, in an ex vivo experiment using fragments of atherosclerotique plaque removed from patients and exposed to modified Fe@C NPs, they have shown a more significant efflux of cholesterol-like substances from atherosclerotic plaques.
Overall, the manuscript is well written.
I do consider that the manuscript may be considered for publication in this journal.
Minor comments:
- I don’t feel that the first images with surgical interventions on rats are really necessary; the same opinion is for the next 2 images showing the carotid endarterectomy and the plaque removed from patient;
- I would suggest to add at least one paragraph in the discussions section in which the authors must emphasize/ discus more about the results obtain from ex vivo experiments and possible mechanism involved.
We tried to take into account these wishes in the article.
Reviewer 3 Report
The article investigates the potential effects of nanoparticles in animal model of atherogenesis. The main concern is represented by the fact that no clear mechanistic insights were discovered, but only association with improvements in the disease. I believe that authors might presents the article as a short comunication or letter instead of research article, since data are only preliminary.
Author Response
Reviewer III
The article investigates the potential effects of nanoparticles in animal model of atherogenesis. The main concern is represented by the fact that no clear mechanistic insights were discovered, but only association with improvements in the disease. I believe that authors might presents the article as a short comunication or letter instead of research article, since data are only preliminary.
Your opinion, as an official expert, is very valuable to us. At the same time, it should be noted that the authors of the article have been searching for the most effective clinical methods and methods for the treatment of atherosclerosis in patients with coronary artery disease for many years. But this article does not consider methods of treating atherosclerosis. It is aimed at finding methods and means of influencing the atherosclerotic plaque itself, in order to find mechanisms that could affect its growth when it is in the lumen of a blood vessel. For this purpose, two experimental models were chosen. One model is aimed at studying the interaction of adipose tissues of an experimental animal with chemically modified nanocomposites (CMN). Of course, this is not an ideal model for research, but we proceeded from the fact that it is adipose tissue that is more involved in the formation of atherogenesis. With the help of morphological research methods, we were able to show that it is CMN, and not the nanomaterial itself in its pure form, that has the property of a special interaction with the adipose tissue of experimental animals. This phenomenon was called by us in the article as the “bordering phenomenon”. In the literature available to us, we did not find such a description of the phenomenon. At the morphological level, this phenomenon can be described as the appearance of an optical clearing zone that appears along the periphery of nanomaterial particles, which is most likely due to an increase in the activity of macrophage cellular elements in the zone located between adipose tissue and CMR. But given that, according to the literature [8 ], it is macrophages that play the leading role in the processes of influencing atherogenesis in particular, we consider this phenomenon as additional possible evidence that hypothetically affects the formation of atherogenesis in general. The second experimental model of the presented article, in our opinion, is an addition to the first model. In this part of the model, on the example of a live atherosclerotic plaque, which was obtained during a planned operation of carotid endarterectomy, it was shown that it is CMNs that have the properties of active interaction with the plaque itself. This phenomenon manifests itself in the form of release of cholesterol-like substances into the physiological solution. Moreover, the yield of these substances directly depends on the applied concentration of modified Fe@C-C18. At 25% concentration of CMN, the level of release of cholesterol-like substances into physiological solution is 3 times higher than at 5% concentration. Thus, the general result obtained, or rather the sequence of searching for these steps, in our opinion, can help other researchers use the described algorithms to achieve more specific problems.
Round 2
Reviewer 1 Report
The results showing fringing phenomenon doesn't show clear evidence of lymphocytes and macrophages. With the histopathological images presented in the manuscript, it is not possible to identify the individual cell types let alone quantify the inflammatory response. The study methods and results depiction needs improvement. The authors can put in more work in showing what type of substances are being released into the saline when the plaque is treated with NPs, a possible way in which the NPs does this and the relevance of this study in assisting to find a potential treatment for atherosclerosis.
Reviewer 3 Report
Sorry, I do not agree with authors' arguments. My previous remarks remain the same and therefore I recommend rejection.